# DsRNA Sequencing for RNA Virus Surveillance Using Human Clinical Samples

**DOI:** 10.3390/v13071310

**Published:** 2021-07-06

**Authors:** Takuma Izumi, Yuhei Morioka, Syun-ichi Urayama, Daisuke Motooka, Tomokazu Tamura, Takahiro Kawagishi, Yuta Kanai, Takeshi Kobayashi, Chikako Ono, Akinari Morinaga, Takahiro Tomiyama, Norifumi Iseda, Yukiko Kosai, Shoichi Inokuchi, Shota Nakamura, Tomohisa Tanaka, Kohji Moriishi, Hiroaki Kariwa, Tomoharu Yoshizumi, Masaki Mori, Yoshiharu Matsuura, Takasuke Fukuhara

**Affiliations:** 1Laboratory of Virus Control, Research Institute for Microbial Diseases, Osaka University, Osaka 565-0871, Japan; m006eb@biken.osaka-u.ac.jp (T.I.); ym6831@biken.osaka-u.ac.jp (Y.M.); ttamura@princeton.edu (T.T.); chikaono@biken.osaka-u.ac.jp (C.O.); 2Department of Surgery and Science, Graduate School of Medical Sciences, Kyushu University, Fukuoka 814-0180, Japan; amori@surg2.med.kyushu-u.ac.jp (A.M.); ttomi@surg2.med.kyushu-u.ac.jp (T.T.); niseda@surg2.med.kyushu-u.ac.jp (N.I.); f_yukiko@surg2.med.kyushu-u.ac.jp (Y.K.); sho_i@surg2.med.kyushu-u.ac.jp (S.I.); yoshizumi.tomoharu.717@m.kyushu-u.ac.jp (T.Y.); m_mori@surg2.med.kyushu-u.ac.jp (M.M.); 3Laboratory of Fungal Interaction and Molecular Biology (Donated by IFO), Department of Life and Environmental Sciences, University of Tsukuba, Ibaraki 305-8577, Japan; urayama.shunichi.gn@u.tsukuba.ac.jp; 4Department of Infection Metagenomics, Research Institute for Microbial Diseases, Osaka University, Osaka 565-0871, Japan; daisukem@biken.osaka-u.ac.jp (D.M.); nshota@gen-info.osaka-u.ac.jp (S.N.); 5Department of Virology, Research Institute for Microbial Diseases, Osaka University, Osaka 565-0871, Japan; tkawagishi@stanford.edu (T.K.); y-kanai@biken.osaka-u.ac.jp (Y.K.); tkobayashi@biken.osaka-u.ac.jp (T.K.); 6Department of Microbiology, Graduate School of Medical Science, Yamanashi University, Yamanashi 400-8510, Japan; tomohisat@yamanashi.ac.jp (T.T.); kmoriishi@yamanashi.ac.jp (K.M.); 7Laboratory of Public Health, Department of Preventive Veterinary Medicine, Division of Veterinary Medicine, Faculty of Veterinary Medicine, Hokkaido University, Hokkaido 060-0808, Japan; kariwa@vetmed.hokudai.ac.jp; 8Department of Microbiology and Immunology, Graduate School of Medicine, Hokkaido University, Hokkaido 060-0808, Japan

**Keywords:** RNA virus, dsRNA, liver transplantation, RNA sequencing

## Abstract

Although viruses infect various organs and are associated with diseases, there may be many unidentified pathogenic viruses. The recent development of next-generation sequencing technologies has facilitated the establishment of an environmental viral metagenomic approach targeting the intracellular viral genome. However, an efficient method for the detection of a viral genome derived from an RNA virus in animal or human samples has not been established. Here, we established a method for the efficient detection of RNA viruses in human clinical samples. We then tested the efficiency of the method compared to other conventional methods by using tissue samples collected from 57 recipients of living donor liver transplantations performed between June 2017 and February 2019 at Kyushu University Hospital. The viral read ratio in human clinical samples was higher by the new method than by the other conventional methods. In addition, the new method correctly identified viral RNA from liver tissues infected with hepatitis C virus. This new technique will be an effective tool for intracellular RNA virus surveillance in human clinical samples and may be useful for the detection of new RNA viruses associated with diseases.

## 1. Introduction

Viruses are intracellular parasites; they can replicate only within cells. They were first identified from plants at the end of nineteenth century [1]. Viruses infect a range of organisms, including humans, animals, bacteria and plants. They have mostly been recognized as pathogens, and they often cause diseases in their infected hosts [2]. In humans, the diseases caused by emerging and reemerging viruses can be life-threatening, as in the cases of the Ebola virus and severe acute respiratory syndrome coronavirus 2 (SARS-CoV-2) [3,4].

Next-generation sequencing technologies have recently been developed and have exerted a strong influence on the field of genomic research. These advanced technologies have enabled us to detect virus genomes more effectively [5]. Most viruses are classified as DNA viruses or RNA viruses, with RNA viruses predominating [6]. As is generally known, the intracellular environment includes many host-derived genomes, such as messenger RNA (mRNA) and ribosomal RNA (rRNA) genomes. At present, it is too difficult to detect exclusively RNA virus genomes efficiently, because the genome size of RNA viruses is small and the volume of the viral genome is very low compared to that of the host genome. To resolve this problem, a method targeting intracellular double-stranded RNA (dsRNA) has been established [7,8,9]. All host-derived RNA forms single-stranded RNA (ssRNA), while almost all ssRNA viruses, including hepatitis C virus (HCV), influenza virus and SARS-CoV-2, form dsRNA in the process of replication [10,11]. Naturally, the dsRNA virus must be detected more efficiently by this method. Therefore, a method for detecting only dsRNA would be more efficient to detect RNA virus genomes. To this end, an environmental viral metagenomic approach targeting intracellular dsRNA has been established in plants and fungi [12,13]. Nevertheless, a method for detection of dsRNA in animal or human samples has not been established. A method for the efficient detection of virus genomes in animal tissues is thus required.

Urayama and colleagues established a dsRNA sequencing method named FLDS (fragmented and primer ligated dsRNA sequencing) to determine the full-length sequences of intracellular RNA viruses. This method includes the following steps: dsRNA purification with a cellulose column, dsRNA fragmentation, adapter ligation to the fragmented dsRNA, cDNA synthesis using a SMARTer system with a primer complementary to the adapter sequence and cDNA amplification. In a previous study, they applied FLDS to marine microorganisms and successfully determined the sequences of several hundred full-length RNA viral segments. However, it is not clear whether the FLDS method is the best method for high-throughput and/or screening application for human samples [14]. In the present study, therefore, we used this FLDS method as the basis for a new method, named the modified dsRNA sequencing method, to detect RNA viruses from human clinical samples. We then examined whether this method is more efficient than the previous approaches for detecting RNA viruses from human clinical samples. 

Living donor liver transplantation (LDLT) is a choice of treatment for chronic or acute liver failure. There are many causes of liver failure, including cholestatic diseases such as primary sclerosing cholangitis (PSC) or primary biliary cholangitis (PBC). In addition, hepatocellular diseases, including liver cirrhosis related to hepatitis B virus (HBV) or hepatitis C virus (HCV) infection, and alcoholic and nonalcoholic fatty liver disease (NASH) are frequent causes [15]. However, in some diseases, including PBC or PSC, the mechanisms underlying the disease development remain to be completely elucidated. It may be possible that viral infections are involved in the development of these and other diseases. Therefore, we investigated whether novel RNA viruses associated with diseases could be identified from liver tissue samples of LDLT recipients using a modified dsRNA sequencing method.

## 2. Materials and Methods

### 2.1. Clinical and Other Samples

The experiments using liver samples from LDLT recipients were conducted at Kyushu University. Between June 2017 and February 2019, we enrolled 57 LDLT recipients at the Kyushu University Hospital. Liver specimens were obtained from each of these recipients (Table 1). We obtained informed consent from all recipients and all studies were approved by the Kyushu University Ethics Committee.

The experiments using lung tissue samples from mice were conducted at Osaka University. Lung samples were prepared from mice using a lethal murine infection model for Pteropine orthoreovirus (PRV) [16]. Four-week-old male C3H/HeNCrl (C3H) mice were purchased from CLEA, Japan. The mice were intranasally infected with 20 μL (4 × 10^5^ PFU) of recombinant strain Miyazaki-Bali/2007 (rsMB) generated by reverse genetics [17]. A lung tissue sample was obtained from the mice infected with rsMB. The mouse experiments were conducted following the approval of the Animal Research Committee of the Research Institute for Microbial Diseases, Osaka University and the guidelines for the Care and Use of Laboratory Animals of the Ministry of Education, Culture, Sports, Science and Technology, Japan. 

The experiments using fungi cells were conducted at Tsukuba University. To obtain fungi cells, mycelial plugs of *Magnaporthe oryzae* strain APU10-199A, containing three RNA viruses, were incubated in YG broth (0.5% yeast extract and 2% glucose) with reciprocal shaking (60 rpm) at 25 °C for two weeks [18]. 

The experiments using liver samples from mice were conducted at Yamanashi University. Five Apodemus speciosus and forty-three Myodes rufocanus bedfordiae were captured in Tobetsu Town, Hokkaido. Liver samples were obtained from each mouse. The mouse experiments were conducted following the approval of the Animal Research Committee of Graduate School of Medical Science, Yamanashi University.

### 2.2. Purification of dsRNA and cDNA Synthesis Using Modified dsRNA-Seq Method

DsRNA was purified as described by Okada et al. with a few modifications [19,20]. A method for intracellular RNA virus surveillance using human clinical samples was established. Based on the FLDS method, this method consists of cellulose column chromatography, fragmentation of dsRNA, cDNA synthesis using random primers and cDNAs derived from dsRNA, and analysis by Hiseq3000 (Figure 1). Briefly, the liver samples were disrupted in phenol/chloroform/isoamyl alcohol (PCI) (Nacalai Tesque) using BioMasher II (Nippi, Tokyo, Japan) and total nucleic acids were manually extracted. Extracted total nucleic acids were purified twice by phenol/chloroform extraction and dsRNA was purified twice through a micro-spin column (empty Bio-spin column; Bio-Rad Laboratories, Hercules, CA, USA) containing cellulose powder (Cellulose D; ADVANTEC, Tokyo, Japan). The remaining rRNA in the elution was removed using RiboMinus Eukaryote System v2 (Invitrogen) according to the manufacturer’s protocol. After additional dsRNA purification using cellulose powder, the dsRNA elution was treated with DNaseI (amplification grade; Invitrogen, Carlsbad, CA, USA) and S1 nuclease (Invitrogen) in nuclease buffer (57 mM CH3COONa, 9.5 mM MgCl_2_, 1.9 mM ZnSO_4_, and 189 mM NaCl) and was then incubated at 37 °C for 15 min. DsRNA was purified using a ZymoClean Gel RNA Recovery Kit (ZymoResearch, Orange, CA, USA). DsRNA for each sample was sheared to about 200 bp using a Covaris S220 ultrasonicator (Covaris). After the denaturation of dsRNA at 95 °C for 3 min, each library was prepared using a SMARTer Stranded Total RNA-Seq Kit v2 - Pico Input Mammalian (TaKaRa/Clontech).

### 2.3. cDNA Synthesis Using the FLDS Method

cDNA was obtained as described previously [14]. In brief, the dsRNA was fragmented using ultrasound and U2 primer was ligated to the 5′-end of fragmented dsRNA. After heat denaturing, cDNA was synthesized using a SMARTer system with U2-comp primer. The cDNA was amplified using KOD-plus Neo (Toyobo, Osaka, Japan) and purified using a 1.25 × SPRIselect reagent kit (Beckman Coulter, Brea, CA, USA).

### 2.4. Total RNA Extraction and cDNA Synthesis for Total RNA Sequencing

Total RNA was isolated using a TRIzol Plus RNA Purification Kit (Invitrogen) according to the manufacturer’s protocol. The RNA fraction was treated with DNase I (amplification grade; Invitrogen) and purified using an RNA Clean & Concentrator-5 (Zymo Research). rRNA was removed using a RiboMinus Eukaryote System v2 (Invitrogen). The cDNA was amplified using a SMARTer Stranded Total RNA-Seq Kit v2 - Pico Input Mammalian (TaKaRa/Clontech).

### 2.5. Illumina Sequencing, Data Assembly and Processing

Paired-end sequencing (100 bp × 2) was performed on the MiSeq and HiSeq3000 platforms. Sequencing reads were trimmed of their adapter sequences by cutadapt 1.8.1 and then mapped onto a custom database, which includes human and microorganism sequences from NCBI-nt by bwa. The taxonomic information of each sequence was assigned and summarized by BioRuby scripts. Trimmed reads were mapped on the viral reference sequences obtained from a public database with CLC Genomics Workbench version 11.0 (CLC Bio, Aarhus, Denmark) using the following parameters: mismatch cost of 2, insertion/deletion cost of 3, length fraction of 0.8 and similarity fraction of 0.8. To detect novel RNA viruses, contigs were obtained de novo exclusively using the CLC Genomics Workbench version 11.0 (CLC Bio) with the following parameters: a minimum contig length of 500, word value set to auto and bubble size set to auto. The contig sequences were compared against the NCBI non-redundant amino acid databases using BLASTX-plus.

### 2.6. Phylogenetic Tree Analysis

Total RNA was extracted from recipient liver samples using a PureLink RNA Mini Kit (Thermo Fisher Scientific, Waltham, MA, USA). The synthesis of the first-stranded cDNA was performed by using a PrimeScript RT Reagent Kit (Takara Bio). The HCV NS5B region sequence was amplified from the synthesized cDNA using PrimeSTAR GXL DNA polymerase (Takara). The following primer set was designed for the NS5B gene: F1, CTACTTGCTCCGAGGAG; R1, AGTCTTCCAGGAGGTCCTTC; F2: GCCGTTAACCACATCAAGTC; R2, ATACCTGGTCATAGCCTCCG; F3, ATCTCAGAAAGCCAGGGGAC; R3, ACCTTTCACAGCTAGCCGTGAC. Amplicons were purified after agarose gel electrophoresis using a Gel/PCR DNA Isolation System (VIOGENE). The NS5B sequence was determined by direct sequencing using a BigDye Terminator v3.1 Cycle Sequencing Kit and ABI3130 Genetic Analyzer (Applied Biosystems). The generated NS5B sequences were aligned using Clustal W. Neighbor-joining phylogenetic trees were inferred in MEGA X using the Tajima–Nei model and gamma-distribution of rates among sites. The confidence of the branches was assessed by the bootstrap test using 500 replicates.

### 2.7. Virus Infection Assay

All cell lines were cultured at 37 °C under the conditions of a humidified atmosphere and 5% CO_2_. The Huh7-derived cell line Huh7.5.1 was kindly provided by F. Chisari, and was maintained in DMEM (Nacalai Tesque, Kyoto, Japan) supplemented with 100 U/mL penicillin, 100 μg/mL streptomycin (Sigma), and 10% fetal bovine serum (FBS). Primary human hepatocytes (PHHs) and PXB cells were isolated from urokinase-type plasminogen activator transgenic/SCID mice inoculated with PHH (PhoenixBio, Hiroshima, Japan) and cultured in the medium according to the manufacture’s protocol. pHH-JFH1 encoding full-length cDNA of the JFH1 strain [21,22] was kindly provided by T. Wakita. pHH-JFH1-E2p7NS2mt, which contains three adaptive mutations in pHH-JFH [23], was introduced into Huh7.5.1 cells/ Then, HCVcc in the supernatant was collected after serial passages, and infectious titers were determined by a focus-forming assay and expressed in focus-forming units (FFU) [22]. Huh7.5.1 cells and PHHs were inoculated with HCVcc and then incubated for three days.

The human hepatoblastoma-derived HepG2-hNTCP-C4 cells were maintained in the above medium supplemented with 400 μg/mL G418 (Nacalai Tesque). HepAD38.7 cells were cultured in DMEM/F-12 medium supplemented with 10% FBS, 100 U/mL penicillin, 100 μg/mL streptomycin, 18 μg/mL hydrocortisone (Sigma), 5 μg/mL insulin (Sigma), 400 μg/mL G418, and 400 ng/mL tetracycline (Nacalai Tesque). HepG2-hNTCP-C4 cells and HepAD38.7 cells were kindly provided by Dr. T. Wakita. The identification of human sodium taurocholate co-transporting polypeptide (hNTCP) as an HBV receptor led to generation of the HepG2-hNTCP-C4 cell line, which stably expresses hNTCP and therefore readily permits the infection and replication of HBV [24,25]. For HBV infection, HepG2-hNTCP-C4 cells were seeded on 12-well plates (Iwaki, Tokyo, Japan) coated with collagen type 1 and incubated overnight. To obtain the HBV-containing culture supernatants, we cultured HepAD38.7 cells with tetracycline-free medium. Then, the HepG2-hNTCP-C4 cells were inoculated with 1000 genome equivalents (GEq)/mL of HBV in the above medium supplemented with 2% DMSO (Sigma) containing 4% PEG 8000 (Nacalai Tesque), and the culture medium was replaced every two days. Extracellular and intracellular HBV DNA were extracted as described previously [26]. Briefly, the cell pellets were lysed by lysis buffer (50 mM Tris-HCl [pH 7.4], 1 mM EDTA, 1% NP-40) at 4 °C for 15 min. After centrifugation at 15,000 rpm for 5 min, the supernatant was incubated with 7 mM magnesium acetate (MgOAc), 0.2 mg/mL of Dnase I (Roche), and 0.1 mg/mL of Rnase A (Sigma) at 37 °C for 3 h. After the addition of 10 mM EDTA, the lysates were digested by proteinase K (0.3 mg/mL; Thermo Fisher Scientific) and 2% SDS at 37 °C for 12 h. Extracted HBV DNA was purified by phenol–chloroform–isoamyl alcohol, precipitated with ethanol, and resolved in pure water. Total RNA was extracted by using a Pure Link RNA Mini Kit (Thermo Fisher Scientific) according to the manufacturer’s protocol. Quantitative PCR (qPCR) for HBV rcDNAs was performed by using Fast SYBR green master mix (Applied Biosystems, Foster City, CA, USA). The following primers were used for the detection of HBV DNA: 5′-GGAGGGATACATAGAG GTTCCTTGA-3′ and 5′-GTTGCCCGTTTGTCCTCTAATTC-3′.

## 3. Results

### 3.1. Comparison between Modified dsRNA-Seq and Other Conventional Methods

We examined whether this modified dsRNA-seq method was more efficient at detecting virus reads than other conventional methods. Three methods were compared—rRNA-depleted total RNA sequencing (total RNA-seq), the FLDS method and our modified dsRNA-seq method—using fungus samples and two LDLT recipient liver samples. In the analysis of the fungus samples, for every 100,000 reads, total RNA-seq detected 4526 reads of viral genome (4.53%), the FLDS method detected 55,132 reads (55.13%) and the modified dsRNA-seq method detected 26,185 reads (26.19%) (Figure 2A). The FLDS method was thus the most effective for detecting viral reads using a fungus sample. On the other hand, the modified dsRNA-seq method was also reasonably effective, detecting five-fold more reads than total RNA-seq. Next, two recipient liver samples were infected with HCV, and the HCV-RNA copies of LDLT recipients in a preoperative blood test were 5.2 and 6.2 log IU/mL. These two liver samples were then used to compare the efficiency of the three methods. For every 100,000 reads, the total RNA-seq method detected 22 reads (0.02%) and 10 reads (0.01%), the FLDS method detected 91 reads (0.09%) and 68 reads (0.07%), and the modified dsRNA-seq method detected 411 reads (0.41%) and 388 reads (0.39%) of viral genome in the two samples, respectively (Figure 2A, Table 2). Most of the virus reads detected from each method were derived from RNA viruses. The modified dsRNA-seq method was the most efficient of the three methods at detecting RNA viruses. Finally, we compared the efficiency of HCV genome detection among the methods. For every 100,000 reads, the FLDS method detected 58 reads (0.06%) and 78 reads (0.08%) and the modified dsRNA seq method detected 362 reads (0.36%) and 410 reads (0.41%) of viral genome in the two samples, respectively (Figure 2B). The total RNA-seq method did not detect HCV reads in either sample. We confirmed that the modified dsRNA-seq method was more efficient for the accurate detection of virus reads than the other conventional methods, with an approximately ten-fold difference compared to the FLDS method for detecting the HCV genome.

### 3.2. Modified dsRNA-Seq Analysis in Animal Tissues and Cultured Cells

Above, we demonstrated that viral genomes were accurately detected in clinical samples by the modified dsRNA-seq method. Next, we evaluated whether this method would work effectively in animal samples or cultured cells. Lung tissue of a reovirus-infected mouse and Huh7.5.1 cells and PHHs inoculated with HCV (JFH-1 strain) were used to test the feasibility of this method. Surprisingly, 91.8% and 60.8% of total reads in the lung tissue from the mouse and the cultured cells were derived from virus reads, respectively (Figure 3A). Reovirus and HCV were detected from the mouse lung tissue and the cell line inoculated with HCV, respectively. The sequence reads were mapped to each viral genome, and they were almost assembled into a full-length genome (Figure 3B,C). We concluded that this method could applied to animal samples and cultured cells, and that it has an advantage in read coverage, with the ability to determine nearly the full-length of the viral genome.

### 3.3. Intracellular RNA Virus Surveillance and Library Construction

We next examined the feasibility of using the modified dsRNA-seq method with human clinical samples. For this purpose, we assessed whether the modified dsRNA-seq method could identify new RNA viruses associated with diseases or clinical states in liver samples obtained from LDLT recipients (n = 57) (Table 1). Three or four liver samples were mixed and then evaluated them with a next-generation sequencer. To analyze the RNA virus surveillance using recipient liver samples, we compared the raw sequence reads generated by NGS against the NCBI nucleotide database, and we constructed a library of the viruses with the greatest number of hits (Table 3). HCV was correctly detected from the recipient liver sample infected with HCV. However, new RNA viruses could not be identified from other liver samples not infected with HCV in this analysis (Table 3). Human endogenous retrovirus K, which was detected in several samples, was considered to be a contaminant from reagents. These results indicated that the throughput could be increased by using mixed samples.

### 3.4. Detection of HCV Genome Using Modified dsRNA-Seq

We could detect HCV correctly from human clinical samples using the modified dsRNA-seq method. In the analysis of recipient liver samples, we used eleven liver samples previously infected with HCV, nine of which were found to contain HCV-RNA in the preoperative blood test (Figure 4A). We could detect HCV reads from preoperative HCV-RNA-positive liver samples using the modified dsRNA-seq method (Figure 4B). Samples No. 8 and 19 had low detection ratios of HCV-RNA at 0.003% and 0.0001%. HCV-RNA was not detected in these samples by preoperative blood tests. These results suggest that even small volumes of HCV-RNA reads could be detected by using this modified dsRNA-seq method. Viral reads obtained from a liver sample (No.13) were mapped to a major HCV genome, and we could determine almost the entire full-length HCV genome, except for a part of the 3′-UTR; the sequence coverage was 98% (Figure 4C). In this analysis, we could detect the HCV genome correctly from HCV-RNA-positive liver samples. In addition, viral reads obtained from this analysis were mapped and the full-length genome could almost be determined with high sequence coverage. Next, for detailed analysis using the viral reads generated by the modified dsRNA-seq data, we compared four contigs obtained by de novo assembly from regions with high coverage of the HCV genome (No.13) and a reference sequence. We performed phylogenetic tree analysis in the NS5B region. The HCV sequence in the NS5B region obtained from a liver sample (No.13) was included in the category of genotype 2a (Figure 4D). Next, we analyzed quasi-species using the obtained HCV reads and evaluated whether there was diversity in high variable region 1 (HVR1). We could obtain five major sequences and many minor sequences (Figure 4E). In sum, by using the modified dsRNA-seq method, we were able to easily perform a detailed analysis using viral reads with high coverage, and to obtain quasi-species data of the viral genome.

### 3.5. Detection of DNA Viral Genome with a Secondary Structure Using Modified dsRNA-Seq

The current modified dsRNA-seq method was established to detect RNA viruses. Interestingly, the method allowed us to identify HBV in an analysis of samples obtained from a recipient liver infected with HBV. In this analysis, we used eight liver samples previously infected with HBV, among which five liver samples had been revealed to contain HBV-DNA in a preoperative blood test (Figure 5A). HBV reads could be detected only in a liver sample rich in viruses (No.11), suggesting that the sensitivity for HBV detection was lower than that for the RNA virus (Figure 5B). HBV viral reads obtained from this liver sample were mapped (Figure 5C). Moreover, we found that the viral contig sequence obtained by de novo assembly from a high coverage region was consistent with the sequence that included the epsilon region. The epsilon region forms a secondary structure that contains a dsRNA structure [27]. Therefore, we concluded that this method could recognize an epsilon region that formed a dsRNA structure in the process of dsRNA extraction. We performed in vitro cell line experiments to examine whether HBV was detected using this method. HepG2-hNTCP-C4 cells were infected with HBV and their lysates were used to evaluate whether HBV could be detected by this method. HBV-DNA copies in the lysates of HepG2-hNTCP-C4 cells were measured and found to be 8.7 log copies/well (Figure 5D). We could detect HBV reads in this analysis, and the HBV viral reads accounted for only 0.02% of the total reads in HepG2-hNTCP-C4 cells (Figure 5E). HBV viral reads obtained by this analysis were mapped and we found that there was high sequence coverage around the epsilon region of the HBV genome as in liver tissue (Figure 5F). These results suggest that the modified dsRNA-seq method was not only an efficient RNA surveillance method, but also could recognize the secondary structure of the viral genome.

### 3.6. Detection of the Fulton Virus in Wild Mus Musculus

RNA virus surveillance was performed using the modified dsRNA sequencing method from liver samples of wild rodents obtained in Hokkaido. Four or five liver samples were mixed and analyzed. We were able to identify the Fulton virus in some liver samples of Myodes rufocanus bedfordiae (Table 4). The Fulton virus consists of three segments (L, M, S) and is a negative-sense RNA virus. This virus was recently reported to have been detected in wild rodents in New York City [28]. It was reported that the Fulton virus was related to *Bunyavirales* families and was associated with liver disease in wild rodents. In this analysis, we could identify the L and M segments of the Fulton virus, and when viral reads obtained from this liver sample were mapped, we found that they gave a high sequence coverage (Figure 6A,B). We could detect the Fulton virus from liver tissues of wild rodents in Hokkaido. We could not demonstrate the association between Fulton virus and any diseases. These results demonstrated that this method was applicable not only to clinical human samples but also to other animal samples.

## 4. Discussion

In the current study, we established a dsRNA-seq method as an RNA virus surveillance method by improving the FLDS method. RNA viruses infected in human clinical samples or animal samples could be detected accurately by using this method. Moreover, we could almost determine the full-length viral genome with high coverage by analyzing viral reads obtained from this method. In addition, HCV could be detected from a recipient liver sample infected with HCV, and we could perform detailed analyses such as phylogenetic tree analysis or quasi-species analysis. We demonstrated that this modified dsRNA-seq method had the potential to detect viral reads efficiently in a wide variety of samples. 

As mentioned above, we developed this method based on the FLDS method. In the FLDS method, cDNA synthesis was performed by ligating loop primers on the 3′-terminal end followed by reverse transcription. This protocol had the advantage that loss of both ends due to PCR was minimized and the full-length genome could be analyzed [14]. In the dsRNA-seq method, we simplified this protocol by annealing random primers instead of loop primers. This enabled high-throughput evaluation to process several samples, so that viral reads could be detected more efficiently—about one thousand times more efficiently than the total RNA-seq method and twenty times more efficiently than the original FLDS method. In FLDS method, U2 primer was ligated to dsRNA fragment and U2-comp primer (complementary to U2 primer) was annealed to start reverse transcription [14]. In modified dsRNA-seq method, random primer was annealed to dsRNA fragment. We consider that ligation efficiency should not be high. This difference may affect the required amount of start material. We assume that simplification of method is key factor for the detection rate of viral reads. According to the removal of ligation step in modified dsRNA-seq, a dsRNA purification step (required for removal of free-U2 adapter) was also removed. This factor may affect viral read ratio. Moreover, our new method efficiently detected HCV from all liver tissues infected with HCV, and it was suggested to have the potential to identify a novel RNA virus from clinical samples. Interestingly, HBV reads were also detected in this analysis, and the contig sequence obtained from HBV reads was consistent with the epsilon region. We considered that it may be possible to recognize the secondary structure of the epsilon region in the process of dsRNA extraction using this method. This modified dsRNA-seq method detected the viral genome, and it may be possible to further apply it as a tool to recognize the secondary structure of the viral genome. Although HBV-DNA was detected by preoperative blood test in five liver samples, HBV viral reads were detected only in a sample (No.11). The sample (No.11) had higher concentration of HBV-DNA. Also, HBV viral reads obtained from HepG2-hNTCP-C4 cells were detected and HBV-DNA copies in the lysate of those cells was high, it was found to be 8.7 log copies/well. This may require the sufficient concentration of HBV-DNA to detect and analyze correctly.

In this analysis, the raw sequence reads generated from NGS were evaluated by BLAST. BLAST compares the similarity between sequences, which means that only known sequences registered in the NCBI database can be evaluated. It is difficult to evaluate unidentified sequences or viruses; this is the limitation of virus surveillance. In our analyses, 30%–40% of unassigned reads were detected from human clinical samples using the modified dsRNA-seq method. It is possible that viruses associated with diseases were present in the unassigned reads. Therefore, it will be necessary to find a better method to search for viruses.

Although HBV and HCV infect the human liver, organs such as the liver have their own immune systems, and thus it may be difficult for other viruses to infect human organs while escaping from their immune systems. Unlike the lungs or intestinal tract, the liver has no opportunity to directly contact the outside environment with its many pathogens. Novel RNA viruses may be identified from clinical samples, such as lung or intestinal tract samples. We should apply this modified dsRNA-seq method to these other human clinical samples because there is the potential to detect viruses associated with diseases of unknown etiology, which could lead to the discovery of new treatments and preventive measures. We analyzed about 60 recipient liver samples in this analysis, which is still a relatively small number. Further analyses will be needed to identify RNA viruses from human clinical liver samples. 

We established the modified dsRNA-seq method to detect RNA viruses more efficiently from clinical samples. We demonstrated that this method detected RNA viruses more efficiently than other conventional methods. In the future, it may be possible to elucidate associations between various viruses and diseases by analyzing the organ-specific viromes obtained by using this method. This method was an effective tool for intracellular RNA virus surveillance using human clinical samples, and it was shown to have good potential for the detection of new RNA viruses.

## Figures and Tables

**Figure 1 viruses-13-01310-f001:**
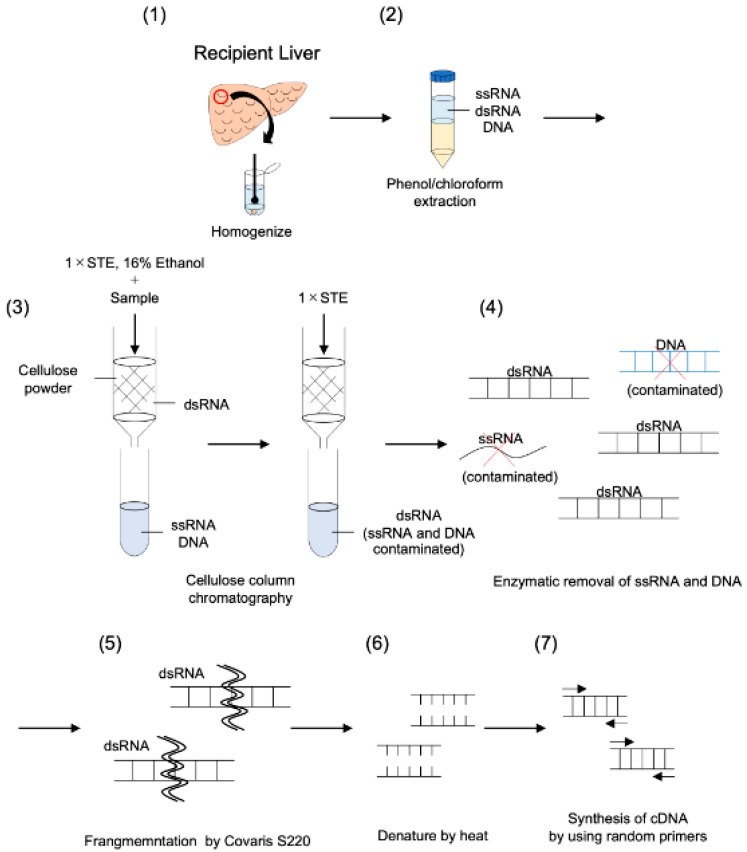
Schematic workflow of the modified dsRNA-seq method. (1) Preparation of liver tissues and homogenization. (2) Extraction of nucleotide contained in dsRNA from clinical samples using phenol/chloroform extraction. (3) Purification of dsRNA using cellulose column chromatography. (4) Enzymatic removal of host-derived DNA and ssRNA from the purified dsRNA. (5) Fragmentation by Covaris S220. (6) Thermal denaturing. (7) Synthesis of cDNA by annealing random primers. cDNAs obtained from this method were analyzed by Hiseq3000. Details of this method are described in the Materials and Methods section.

**Figure 2 viruses-13-01310-f002:**
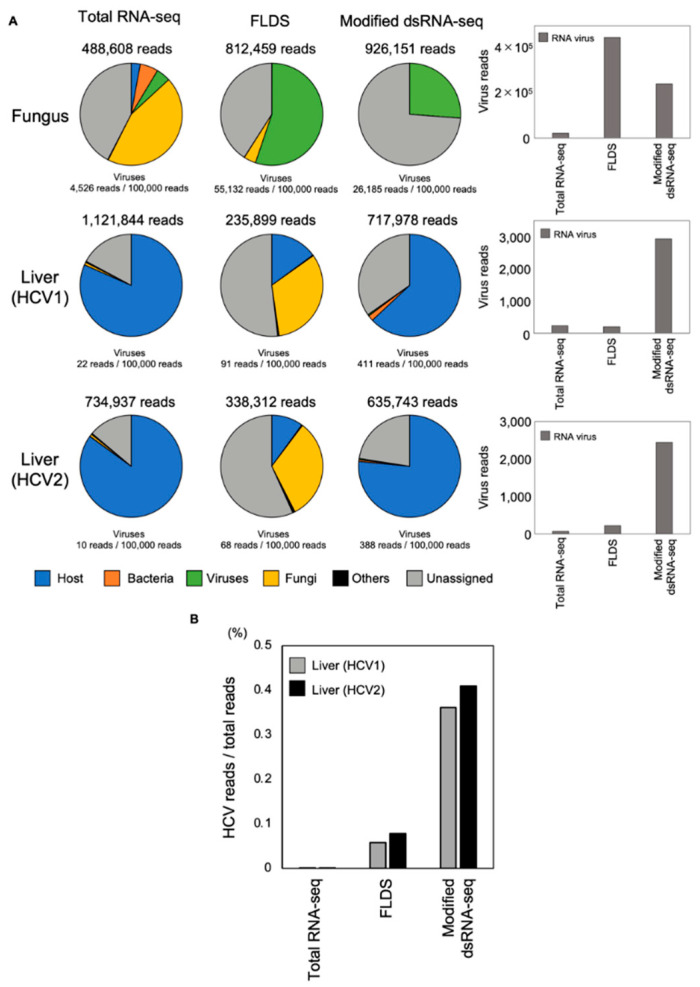
Evaluation of the utility of the modified dsRNA-seq method. Samples from a fungus and two recipient livers with liver cirrhosis were analyzed by the total RNA-seq, FLDS and modified dsRNA-seq methods. (**A**) Fungus and liver samples (HCV1 and HCV2) are compared in a pie chart. Blue represents host-derived genome reads, orange Bacteria, green Viruses, yellow Fungi, black Others, and gray Unassigned reads. Virus reads detected by each method are shown in the bar graph at right. Gray represents RNA virus reads. (**B**) Comparison of the read number of HCV genome detected by each method. The gray bar represents HCV1 in the liver and the black bar represents HCV2 in the Liver.

**Figure 3 viruses-13-01310-f003:**
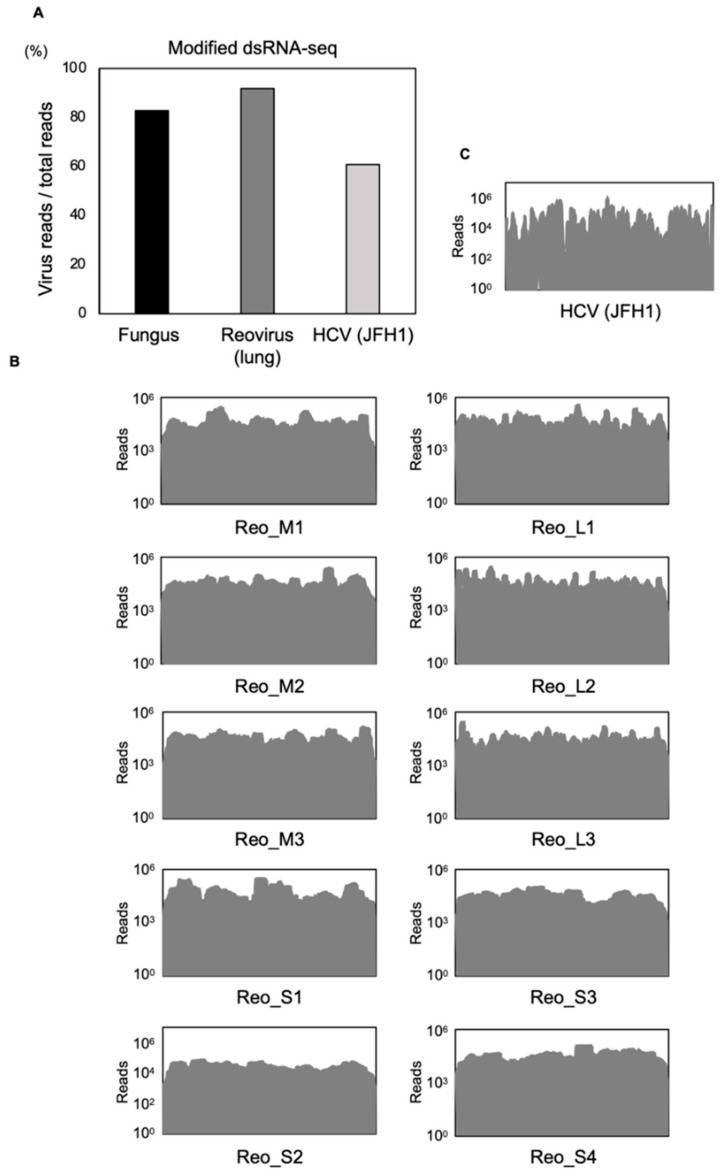
Analysis of cell line, animal tissues or environmental samples using the modified dsRNA-seq method. (**A**) Comparison of mapped viral read frequencies among the three methods using a fungus naturally infected with MoCV1, mouse lung tissue infected with reovirus and Huh7.5.1 cells and PHHs infected with HCV (JFH-1). (**B**,**C**) Genomic coverage of each viral segment from the modified dsRNA-seq analysis. Reo_M; Reo_medium genome segment, Reo_S; Reo_small genome segment, Reo_L; Reo_large genome segment.

**Figure 4 viruses-13-01310-f004:**
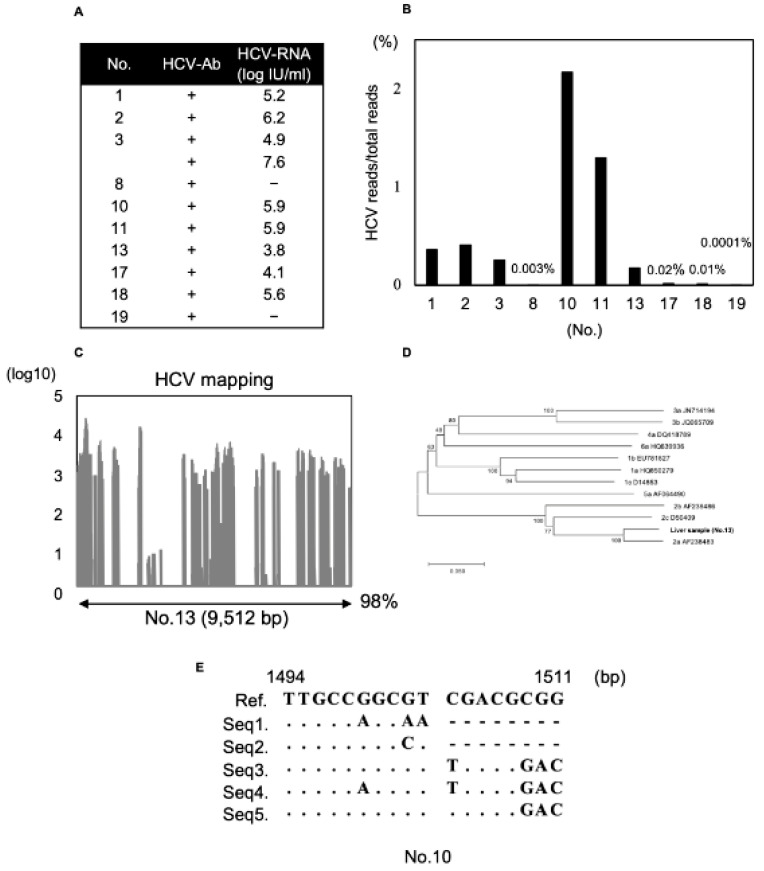
Detection of HCV reads obtained from recipient liver samples. (**A**) Recipient liver sample list of HCV infection previously with HCV-Ab positive, HCV-RNA was evaluated by preoperative blood test. (**B**) HCV reads compared to total reads obtained from each liver sample were shown in bar graph. (**C**) HCV reads from a liver sample (No.13) were mapped. A bar graph is shown and the percentage of the total number of reads that were mapped to the HCV genome is indicated. (**D**) Phylogenetic tree analysis in the NS5B region by the viral sequence obtained from HCV reads (HCV No. 13). (**E**) Analysis of quasi-species using HCV reads obtained from liver samples (No.10). Alignment of HVR1 (1494–1511 bp). Nucleotide positions are numbered according to GenBank accession number NC004102. The five major sequences are listed.

**Figure 5 viruses-13-01310-f005:**
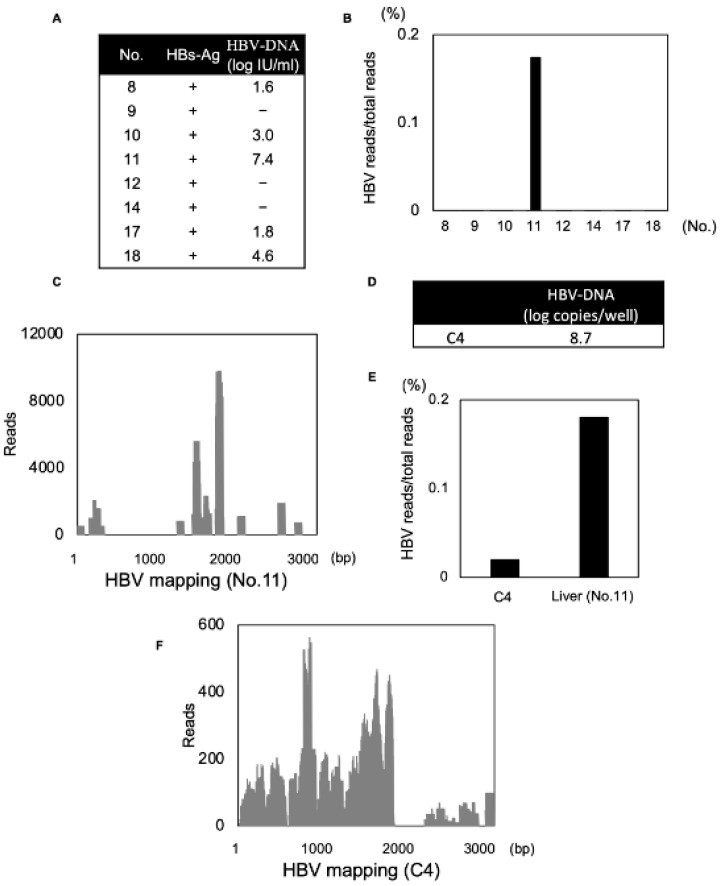
Detection of HBV. (**A**) Sample list of HBsAg positive recipients. (**B**) HBV reads compared to total reads obtained from each liver sample are shown in a bar graph. (**C**) HBV reads identified from a liver sample were mapped to the HBV genome and bar graph was indicated. (**D**,**E**) HBV-DNA copies and HBV reads compared to total reads obtained from the lysates of HepG2-hNTCP C4 cells are shown. (**F**) HBV reads identified from in vitro cell line experiments were mapped to the HBV genome and bar graph was indicated.

**Figure 6 viruses-13-01310-f006:**
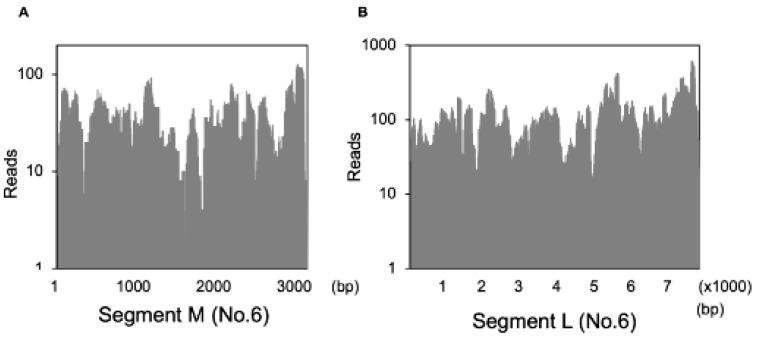
Detection of the Fulton virus. (**A**,**B**) Fulton viral reads identified from liver tissues of Myodes rufocanus bedfordiae (No.6) were mapped to the L and M segment of the Fulton virus and bar graph was indicated.

**Table 1 viruses-13-01310-t001:** Etiology of each patient.

No.	Disease	No.	Disease
1	HCC (HCV)	12	Wilson disease
2	LC (HCV)		PBC
3	HCC (HCV)		AIH
	HCC (HCV)		LC (HBV), HCC
4	NASH	13	AIH
	NASH, HCC		NASH
5	AIH, PBC		LC (HCV)
	AIH, PBC		NASH
6	Calori disease	14	LC (Alcohol)
	PSC		LC (Alcohol), HCC
7	VOD		NASH
	LC (Alcohol)		LC (HBV), HCC
	Wilson disease	15	LC, HCC
	LC (nonBnonC)		LC (Alcohol)
8	LC (HCV)		NASH
	LC (nonBnonC)	16	Acute liver failure
	PBC		LC (Alcohol)
	HCC (HBV)		PBC
9	LC (Alcohol)	17	LC (HCV)
	Acute liver failure (HBV)		HBV
	LC (Alcohol)		PBC
	AIH, HCC	18	Acute liver failure (HCV)
10	Acute liver failure (HBV)		LC (Alcohol)
	LC (nonBnonC), HCC		LC (HBV)
	LC (HCV)	19	LC (HCV)
	PBC		AIH, PSC
11	LC (nonBnonC), HCC		LC (Alcohol)
	LC (HBV)		
	LC (Alcohol)		
	LC (HCV), HCC		

HCC: hepatocellular carcinoma; LC: liver cirrhosis; NASH: nonalcoholic steato-hepatitis; AIH: autoimmune hepatitis; PBC: primary biliary cholangitis; PSC: primary sclerosing cholangitis; VOD: hepatic veno-occlusive disease.

**Table 2 viruses-13-01310-t002:** Comparison of total RNA-seq method, FLDS method and modified dsRNA-seq using fungus sample and two recipient liver samples infected with HCV.

	Total Reads	Host	Bacteria	Viruses	Fungi	Others	Unassigned
Total RNA-seq							
Fungus	488,608	14,119 (2.9%)	28,056 (5.7%)	22,115 (4.5%)	216,259 (44.3%)	956 (0.2%)	207,103 (42.4%)
Liver (HCV1)	1,121,844	915,137 (81.6%)	166 (0.0%)	247 (0.0%)	8862 (0.8%)	4482 (0.4%)	192,950 (17.2%)
Liver (HCV2)	734,937	624,222 (84.9%)	610 (0.1%)	73 (0.0%)	5275 (0.7%)	2752 (0.4%)	102,005 (13.9%)
FLDS							
Fungus	812,459	553 (0.1%)	44 (0.0%)	447,923 (55.1%)	30,944 (3.8%)	586 (0.1%)	332,409 (40.9%)
Liver (HCV1)	235,899	35,165 (14.9%)	376 (0.2%)	214 (0.1%)	76,749 (32.5%)	1071 (0.5%)	122,324 (51.9%)
Liver (HCV2)	338,312	33,828 (10.0%)	743 (0.2%)	229 (0.1%)	108,961 (32.2%)	2588 (0.8%)	191,963 (56.7%)
Modified dsRNA-seq							
Fungus	926,151	67 (0.0%)	31 (0.0%)	242,514 (26.2%)	12 (0.0%)	2 (0.0%)	683,525 (73.8%)
Liver (HCV1)	717,978	452,549 (63.0%)	12,419 (1.7%)	2953 (0.4%)	358 (0.0%)	93 (0.0%)	249,606 (34.8%)
Liver (HCV2)	635,743	486,372 (76.5%)	3264 (0.5%)	2464 (0.4%)	196 (0.0%)	117 (0.0%)	143,330 (22.5%)

**Table 3 viruses-13-01310-t003:** List of top hit virus identified from recipient liver samples using modified dsRNA-seq method.

No.	Total	Viruses	Ratio	Virus (Reads)	
	(Reads)	(Reads)	(%)	Top Hit	
1	717,978	2953	0.4	Hepatitis C virus	2889
2	635,743	2464	0.4	Hepatitis C virus	2430
3	556,921	10,956	2.0	Brome mosaic virus	8927
				Hepatitis C virus	2011
4	609,020	43	0.0	Human endogenous retrovirus K	35
5	595,735	13	0.0	Human endogenous retrovirus K	12
6	563,170	38	0.0	Human endogenous retrovirus K	32
7	17,634,237	960	0.0	Human endogenous retrovirus K	482
8	15,890,244	2659	0.0	Human endogenous retrovirus K	2081
9	17,626,711	1091	0.0	Human endogenous retrovirus K	1064
10	12,808,763	341,827	2.7	Hepatitis C virus	341,738
11	14,061,933	220,722	1.6	Hepatitis C virus	195,642
				Hepatitis B virus	24,960
12	18,478,904	636	0.0	Bell pepper endornavirus	272
13	16,294,029	8303	0.1	Hepatitis C virus	7298
14	16,097,692	1085	0.0	Human endogenous retrovirus K	1076
15	12,106,817	1273	0.0	Human endogenous retrovirus K	568
16	15,198,287	764	0.0	Human endogenous retrovirus K	392
17	20,526,142	3935	0.0	Hepatitis C virus	2628
18	15,195,371	2641	0.0	Hepatitis C virus	2347
19	19,094,227	1260	0.0	Human endogenous retrovirus K	862

**Table 4 viruses-13-01310-t004:** List of wild Mus musculus and identification of Fulton virus.

No.	Type of Wild Rodents	Sample	Organ	Identification
1	Apodemus speciosus	5	liver	
2	Myodes rufocanus bedfordiae	5	liver	
3	Myodes rufocanus bedfordiae	5	liver	Fulton virus (L, M)
4	Myodes rufocanus bedfordiae	5	liver	Fulton virus (L, M)
5	Myodes rufocanus bedfordiae	5	liver	
6	Myodes rufocanus bedfordiae	5	liver	Fulton virus (L, M)
7	Myodes rufocanus bedfordiae	5	liver	
8	Myodes rufocanus bedfordiae	5	liver	
9	Myodes rufocanus bedfordiae	4	liver	

## Data Availability

Not applicable.

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
