# Peer review of "DsRNA Sequencing for RNA Virus Surveillance Using Human Clinical Samples"

_viruses, 2021, doi:10.3390/v13071310_

Round 1

Reviewer 1 Report

The author established the modified dsRNA-seq method to detect RNA viruses more efficiently from clinical samples. They demonstrated that this method detected RNA viruses more efficiently than other conventional methods. In the future, it may be possible to elucidate associations between various viruses and diseases by analyzing the organ-specific viromes obtained by using this method. This method was an effective tool for intracellular RNA virus surveillance using human clinical samples, and it was shown to have good potential for the detection of new RNA viruses. Albeit, I consider these findings to provide insight into the field of viruses detection. I still have minor suggestions for the authors.

1, In the current study, the paired-end sequencing (100 bp x 2) was performed on the MiSeq and HiSeq3000 platforms. Is this possible that the author can share and upload these NGS data via  NCBI Gene Expression Omnibus (GEO) system?

2, The author need to check the format of the Reference ( especially for reference 18), I suggest that they can take a look at the recent paper in Viruses  as below

https://www.mdpi.com/1999-4915/13/5/929/htm  

Author Response

Thank you very much for your positive comments. We revised our manuscript in line with your comments. Our point-by-point responses are as follows.

Reviewer 2 Report

The author’s aim is to discover new viruses that cause diseases using human samples or other samples in modified dsRNA-seq method. First, they established the new method named modified dsRNA-seq method to search for RNA viruses and they showed that it is more effective than other conventional methods. In addition, they showed that the new method is able to identify the viral genome and analyze identified viral genome. Although the author failed to identify new viruses cause disease in this study, it is an interesting method. Finally, they showed that new method analyzed not only human samples but also animal samples, such as mouse liver infected with Fulton virus. This may allow us to develop experiments related to virus search in various fields in the future. This manuscript is well-written and informative for developing the new method to search for RNA viruses. There still remains some criticisms before acceptance for publication.

Major comments

  1. The major differences between FLDS method and modified dsRNA-seq is the type of primer. FLDS method uses loop primer (U2 primer) and modified dsRNA-seq method uses random primer. The author showed that the detection rate of modified dsRNA-seq method was approximately ten-fold higher than FLDS method for detecting the HCV genome in Figure 2B. Why is the detection rate of viral reads different depending on the primer?
  2. HBV viral reads are detected only in sample No.11, not in other samples in Figure 5B. However, HBV-DNA is detected in preoperative blood tests in several samples. The author should explain this discrepancy.
  3. The author showed that Fulton virus was identified from mouse liver. If the author has the data of association with Fulton virus and diseases, the author should explain these points in Result or Discussion section.

Minor points

  1. Page 3, Table 1, 14. What does LC-B mean? Is LC-B LC (HBV)? In 18, the authors describe LC (HBV). Use the same terminology as others.
  2. 2 and Table 2. The numbers of total reads for FLDS and modified dsRNA-seq are inconsistent between Fig.2 and Table 2. FLDS has 812459, 235899, and 338312 reads for Fungus, Liver (HCV1), and Liver (HCV2), respectively, in Fig.2, whereas modified ds RNA-seq has 812459, 235899, and 338312 reads for Fungus, Liver (HCV1), and Liver (HCV2), respectively. There may be a mistake. Which is correct, Fig.2 or Table 2 ?
  3. 3. The authors should add more explanations for the Reo_M1, Reo_L1, etc ?

For example, M: middle genome segments.

  1. Page 11, line 9. ‘However, new RNA viruses could not be identified from other liver samples not infected with HCV in this analysis.’ Which No. of the samples showed the result for this sentence in Table 3 ?
  2. Page 14, line 5. Table 6 → Table 4 ?

Author Response

(The authors gave the same response as above.)
